# Effect of the Addition of Fique Bagasse Cellulose Nanoparticles on the Mechanical and Structural Properties of Plastic Flexible Films from Cassava Starch

**DOI:** 10.3390/polym15194003

**Published:** 2023-10-05

**Authors:** Jhon Jairo Palechor-Trochez, Adriana Rocio Chantre-López, Eduardo Argote-Ortiz, Héctor Samuel Villada-Castillo, Jose Fernando Solanilla-Duque

**Affiliations:** Faculty of Agrarian Sciences, Universidad del Cauca, Popayán 190003, Cauca, Colombia; cadriana@unicauca.edu.co (A.R.C.-L.); eduargote_2402@unicauca.edu.co (E.A.-O.); villada@unicauca.edu.co (H.S.V.-C.);

**Keywords:** lignocellulosic biomass, cellulose, nanoparticles, acid hydrolysis, bionanocomposite, mechanical properties

## Abstract

One of the activities most representative of the agricultural sector in Colombia is the production of biodegradable fique fiber. The efficiency of the defiberization process of the fique leaves is very low since a mere 4% of the total weight of the leaf (*cabuya*) is used and marketed. The remaining 96%, composed of fique juice and bagasse, is considered to be waste and discarded, impacting the environment. The aim of this work was to study fique bagasse as a source of cellulose nanoparticles (CNCs). CNCs were obtained by acid hydrolysis and added at 10% to films made from cassava thermoplastic starch (TPS) by the casting method. Structural changes in the CNCs, TPS, and their mixtures were characterized by FTIR-ATR and their morphology and particle size by SEM and TEM microscopy, respectively. Thermal properties were analyzed using DSC and TGA, along with their effect on mechanical properties. Changes in the FTIR spectra indicated that the chemical method adequately removed hemicellulose and lignin from the fiber surface of fique bagasse. The CNCs showed a diameter and length of 7.5 ± 3.9 and 52.7 ± 18.1 nm, respectively, and TPS 10% CNC obtained an increase in mechanical strength of 116%. The obtainment of CNCs from lignocellulosic materials can thus be viewed as a favorable option for the subsequent reinforcement of a polymeric matrix.

## 1. Introduction

The world today is facing an environmental problem of considerable magnitude due to the indiscriminate use of plastics of petrochemical origin. Because their use period is short and they are quickly converted into waste, this kind of material accumulates in landfills. These materials are discarded before much time has elapsed, subsequently taking years for their breakdown in environmental conditions of temperature, humidity, UV rays, etc. [1]. As a consequence of this accumulation, greenhouse gases are produced, with an average emission of 2.27 tons of CO_2_ for one ton of plastic [2,3,4,5]. This amount of greenhouse gases accumulates in the earth’s atmospheric layer, trapping heat in the atmosphere and contributing to global warming. For this reason, the use of materials with biodegradable character is an opportunity to mitigate the environmental contamination derived from the use of plastic materials [6,7,8].

In light of this, much research is focused on developing new plastic materials that are friendly to the environment. These environmentally friendly materials are generally made from renewable sources and are biodegradable under suitable conditions, with a period of decomposition of around 180 days, a phenomenon that occurs by the action of a community of microorganisms. As such, researchers have focused their efforts on seeking natural polymers (proteins, polysaccharides) from biomass, other polymers produced by chemical synthesis (PLA, biopolyester), and yet others from microorganism synthesis (xanthan, poly hydroxybutyrate) [9,10].

The most widely used polymer from a renewable source is starch. It is perhaps the most widely studied at a scientific level because it is a low-cost material, abundant in nature, from a biological base, and biodegradable. Moreover, 75% of this polymer is used to manufacture containers and packages. The polymer can be blended with a plasticizer such as glycerol or sorbitol, among others, forming a material named thermoplastic starch (TPS), but TPS has several disadvantages—principally, its poor mechanical and barrier properties. These poor properties are caused especially by its high water sensitivity due to its hydrophilic character [11,12,13,14].

A solution to these problems could be using different strategies to obtain better mechanical and barrier properties. Today, the highlight used the incorporation of nanoparticles to produce a material known as bionanocomposite [15,16]. The principal characteristic of this kind of material is the blend between a biopolymeric matrix and fibers or particles with dimensions on a nanometric scale [17,18,19]. This kind of blend is used to manufacture material with better mechanical and barrier properties, whether compared with a conventional or pure polymer [15,20,21,22,23]. These properties show some increase in the Young’s modulus, maximum stress, the water barrier, resistance heater, and decrement in the gas’s permeability. In this context, when a composite includes 1 wt.% of cellulose nanoparticles, its maximum stress goes up to approximately 61%, increasing from 320 J/m^2^ to 517 J/m^2^ [24]. This is an argument indicating that the use of nanometric scale particles is open to new alternatives for finding not only better mechanical properties but also a better cost–benefit relationship [16,18].

In principle, any cellulose biomass may be used as a potential source of cellulose from a nanoparticulate material, including crops, waste crops, and agro-industrial byproducts [25]. In this set of options appears fique bagasse, a lignocellulose waste derivative from the fique fiber productive chain [1]. Fique (*Furcraea* spp.) is a plant with 30 wt.% of fibrillates and 70 wt.% of vegetal pulp with high storage of cellulose [26]. This cellulose can be extracted by mechanical and chemical methods or a combination of both [22,27]. In Colombia, fique crops are widely grown, especially in the departments of Cauca, Nariño, Santander, and Antioquia. According to the Ministry of Agriculture and Rural Development [28], in 2020, the area sown was 15.970 ha with a production of 19.703 t, generating around 13.792 t of waste bagasse that requires an alternative solution to prevent environmental pollution and add value. One of these alternatives is the extraction of nanoparticles.

As such, the objective of this paper is the production of cellulose nanoparticles from fique bagasse [1] as an alternative for making integral use of the byproducts of the fique crops, using methods such as acid hydrolysis to obtain cellulose nanoparticles (CNCs) and incorporate them in a starch-based bionanocomposite polymeric matrix to increase the mechanical and barrier properties [29,30].

## 2. Materials and Methods

### 2.1. Materials

Fique bagasse (FB) from the *Uña de Aguila* variety (*Furcraea macrophylla*) was collected in a rural area of the municipality of Totoró in Cauca, Colombia. The FB was washed and dried at 50 °C in a mechanical convection oven (Memmert Gmbh UNB 500, Büchenbach, Germany) at a humidity of 5%. The FB then underwent size reduction to 500 µm using an analytic mill (POLIMIX^®^ PX-MFC 90D, Kinematica AG, Malters, Switzerland). Finally, the FB was heat treated at 125 °C for approximately 2 h to obtain autoclave product (ABC). The reagents used were sodium hydroxide, 15% hypochlorite, and 98% sulphuric acid.

### 2.2. Production of Cellulose Nanoparticles (CNCs)

Following the method described by Kazharska, et al. [31], the CNCs were isolated from the FB, with some modifications in the process according to Yadav et al. [32] and Dasan et al. [33]. The FB treated previously was placed in an alkaline treatment using a 10% NaOH solution for approximately 2 h in a bioreactor, with constant shaking, at 120 rpm (CENTRICOL, Medellín, Colombia), equipped with a temperature control system (LAUDA ALPHA RA 8, Lauda-Königshofen, Germany) adjusted to 80 °C. Then, the FB was blanched in a 10% NaClO solution for 3 h to 30 °C. The sample obtained was washed with distilled water in a centrifuge at 13,000 rpm (HERMLE Labortechnik Z 306, Wehingen, Germany) to lower its pH to 6. After that, the blanched FB was dried at a temperature of 50 °C to an absolute humidity of 4% and finally underwent a hydrolysis process using a 10% H_2_SO_4_ solution at 70 °C for 7 h.

### 2.3. High-Resolution Optical Microscopy

The particle size of the FB and hydrolyzed FB (HFB) was measured using a high-resolution optical microscope (NIKO MICROPHOT, Tokyo, Japan) coupled to a digital camera (Nikon DS-2Mv 2Mp, Tokyo, Japan). Three photos of each sample were taken at a 4× magnification, using Lugol as a staining agent to contrast the image. The photographs were analyzed using digital image analyzing software (Image-Pro Plus; Media Cybernetics, Silver Spring, MD, USA), measuring the maximum and minimum radius and area.

### 2.4. Electron Transmission Microscopy (TEM)

The hydrolyzed bagasse cellulose (FB) samples were prepared by suspending 5 mg in 1 mL of deionized water to disperse the agglomerated particles for 5 min using an ultra-sonification bath (Bransonic 3510E-MTH, output 42 kHz, Branson Ultrasonics Co., Danbury, CT, USA). Then, one drop of the aqueous suspension was placed onto a copper TEM grid with a carbon membrane (Ted Pella^®^, Redding, CA, USA). Next, one drop of uranyl acetate was applied to each grid for 10 min to stain the samples. The images were obtained in a TEM equipped with a photographic camera CCD to 80 kV at 40.

### 2.5. Color Properties

The color determination employed a colorimeter (CM-5, Konica Minolta Sensing Inc., Osaka, Japan), with a D65 light source, an angle of observation of 10°, and a calibration standard Z (89.5); X (0.3176); Y (0.3347). A black background was used for measuring the samples, with three bursts of light for each sample. The CIEL*a*b* color scale was used. 

The colorimetric indexes were calculated using the following equations:

Yellow Index (*YI*)
(1)$$YI=\frac{142.6\times b^{*}}{L^{*}}$$
whiteness index (*WI*), saturation index (*SI*) and brown index (*BI*)
(2)$$WI=100-\sqrt{{\left(100-L^{*}\right)}^{2}+{a^{*}}^{2}+{b^{*}}^{2}}$$
(3)$$SI=\sqrt[{}]{{a^{*}}^{2}+{b^{*}}^{2}}$$
(4)$$BI=\frac{100\times (x-0.31)}{0.17}$$
where:(5)$$x=\frac{\left(a^{*}+1.75\times L^{*}\right)}{\left(5.645\times L^{*}+a^{*}-3.012\times b^{*}\right)}$$

Color difference (Δ*E***_ab_*)
(6)$$∆E_{ab}^{*}={\left[{\left({∆L}^{*}\right)}^{2}+{\left({∆a}^{*}\right)}^{2}+{\left({∆b}^{*}\right)}^{2}\right]}^{0.5}$$
where: $$∆L^{*}=L_{muestra}^{*}-L_{estándard}^{*}$$
$$∆a^{*}=a_{muestra}^{*}-a_{estándard}^{*}$$
$$∆b^{*}=b_{muestra}^{*}-b_{estándard}^{*}$$

Negative CI values between −40 to −20 related to colors that go from blue-purple to deep green colors. From −20 to −2 are deep green to green-yellow colors. From −2 to +2 are yellow-green colors. From +2 to +20, pale yellow to intense orange colors. And from +20 to +40, intense orange to deep red colors [34].

### 2.6. Fourier Transform Infrared Spectroscopy (FTIR)

The FTIR spectra were obtained in an IK-Fourier Shimadzu IRAffinity-1S spectrometer (Shimadzu, Inc., Columbia, MD, USA) with a deuterated triglycine sulfate (DTGS) detector. Particles from FB and films with CNC were studied via horizontal attenuated total reflectance (ATR) sampling accessory (ATR- MIRacle 10 Shimadzu, Inc., Columbia, MD, USA) with a diamond single-reflection crystal at a 45° angle of incidence. The spectrum of each sample was obtained by taking the average of 45 scans at a resolution of 4 cm^−1^ at 25 °C. Happ–Genzel apodization was used, with magnitude phase correction. A background spectrum was recorded in air (without sample) prior to each spectrum measurement. Spectra were acquired between 600 and 4000 cm^−1^, and the average of the triplicate for each sample was reported. Spectral analysis was performed using Excel software. Spectra were both baseline-corrected and normalized (between 0 and 1) [35].

### 2.7. Film Preparation

The bionanocomposite matrix was obtained by the casting method, blending 30 wt.% of cassava starch, plasticizer (glycerol in a proportion of 30 wt.% compared to the starch) and water 70 wt.%. The components were mixed with mechanical stirring in a water bath at 78 °C. Then the mix was poured into a smooth tray and placed in a convection oven at 50 °C for 24 h. The same procedure was employed for the incorporation of the NPCs, using for their dispersion an Ultra-Turrax (IKA^®^ T25 easy clean digital, Breisgau, Germany) at 12,000 rpm [36].

### 2.8. Thermal Properties

#### 2.8.1. Differential Scanning Calorimetry (DCS)

The thermal properties of the starches were determined by differential scanning calorimetry (DSC) (Q20 TA Instruments, New Castle, DE, USA) under an air flow at 100 mL min^−1^. The thermal parameters were recorded in duplicate. A TPS film sample with CNCs and without CNCs (4 to 5 mg, dry weight basis) was placed in a 40 μL aluminum pan, and distilled water was added to give a sample to water weight ratio of 1:3. The pan was sealed, and the sample was allowed to equilibrate overnight at 4 °C before analysis. In the DSC, the sample was kept at 25 °C for 1 min, followed by heating from 25 to 95 °C at a rate of 10 °C/min. Temperatures of onset of gelatinization (To), peak (Tp), and endset (Te), together with enthalpy of gelatinization (ΔH), were recorded in duplicate [23]. In order to obtain the DSC curves, the instrument was calibrated with indium, 99% purity, m.p. = 156.6 °C, DH (Nominal) = 28.71 J/g, DH (Registered) = 28.58 J/g [37].

#### 2.8.2. Thermogravimetric Analysis (TGA)

In this test, samples of TPS film with CNCs and without CNCs were used. ASTM standard E1131-08 (2014) was used for the analysis of heat degradation of the samples. These were previously prepared for a minimum period of 48 h at 23 ± 1 °C and 50 ± 10% RH. In all the experiments, quantities of 5 mg of the material were heated from 30 to 600 °C, at a rate of 20 °C/min that was kept constant during the experimental phase, under an inert atmosphere with a nitrogen flow rate of 30 mL/min using thermogravimetric analysis equipment (Model Q50, TA Instruments, New Castle, DE, USA) [35].

### 2.9. Evaluation of Mechanical Properties

The tensile properties of the flexible films obtained were measured: Young’s modulus (MPa), tensile stress (N), and elongation (%). The samples were taken to a temperature chamber (Binder KBF 115, Bohemia, NY, USA), where they were stored under constant conditions of relative humidity (50%) and temperature (23 °C) for 8 days. Universal test equipment was used (Shimadzu model EZ-L) following the ASTM D882-10 standard, which establishes the procedure for carrying out tensile testing on films. The following operating conditions were used: a 500 N cell, 50 mm/min spindle speed, data collection rate of 500 points/s, and 50 mm distance between vices. Samples were cut in longitudinal and transversal directions with dimensions of 90 mm length by 20 mm width. Sample thicknesses were taken using a micrometer (Testing Machine, Inc., New Castle, DE, USA, 549) [38].

### 2.10. Bionanocomposite Matrix Barrier Properties

Oxygen permeability was measured employing three circular samples 80 mm in diameter. These were placed in the gas permeability machine (Gas Permeability Tester Perme VAC-VBS, Labthink, Medford, MA, USA) in a gas transmission cell, forming a semi-barrier between the two chambers. The lower chamber was brought to a stable vacuum pressure for 3 h. Then, the gas was fed through the upper chamber at a higher pressure and allowed to pass through the circular sample. The pressure differential ensures that the gas permeates through the sample from the upper chamber to the lower chamber [39].

### 2.11. Scanning Electron Microscopy (SEM)

The samples of film with FB CNCs and without FB CNCs were covered with gold-palladium using a Mini Sputter (SC7620 Quorum Technologies, Lewes, UK) for a time period of 6 min in a pressure vacuum of 10^−4^ mbar. The microstructure and elemental analysis of the samples were examined in an SEM (Zeiss EVO 10, Carl Zeiss Inc., Cambridge, UK) equipped with a lanthanum hexaborite filament and operated at 20 kV and 4000X. SEM images were obtained at an acceleration voltage of 20 kV and an 8.5 mm working distance. Different magnifications were used under Secondary Electron Detector (SE) and Backscattered Electron Detector (BSE) modes. Micrographs were taken at a scale of 10 µm and 20 µm for micrographs of the surface and cross-sections, respectively.

### 2.12. Statistical Analysis

The results obtained using high-resolution optical microscopy for the mechanical and barrier properties were evaluated using ANOVA analysis with a Tukey’s range test at *p*-value < 0.05.

## 3. Results and Discussion

### 3.1. Variation of Particle Size in the Acid Hydrolysis Process

Figure 1 shows the particle size distribution of the FB, and these results reveal that 22.84% of the particles are 0.165 mm in diameter. Most of the mass fractions, meanwhile, are from 0.363 to 0.165 mm in diameter, for a total fraction of 64.87%.

Following the heat and chemical treatments to which the BC was subjected, the maximum and minimum diameter and the area of the particles derived from the hydrolysis process were measured, as shown in Figure 2. With these data, it was observed that the factor with the greatest influence over particle size was temperature. Furthermore, with the combination of temperature, time, and acid concentration, particle size was affected significantly. The change in particle size is evidence that the size and other morphologic properties are affected by the reaction temperature, the processing time, and the concentration of acid in the hydrolysis process [40]. Accordingly, the process conditions were established at a temperature of 70 °C, a sulfuric acid concentration of 10%, and a hydrolysis time of 7 h, with an average area of 61.69 mm, a maximum diameter of 8.73 mm, and a minimum diameter of 5.29 mm. Acid hydrolysis reduced the size of the BC from an average of 165 µm to 8.73 µm in diameter. This is an effect of the hydrolysis of the amorphous zones of the large chains of the cellulose because they are reduced in the crystalline regions of the cellulose that are most resistant to chemical attack, producing particles with a rod form [41].

The observed particle size was not homogeneous due to the presence of agglomerations (see Figure 2), a behavior that may very possibly be an effect of the cellulose drying process because the molecular contact between the CNCs increases through the resultant forces of the removal of the water and the high temperature [42,43]. Other causes might be the high surface potential of the remaining microfibrils and the concentration of acid that affects the amorphous region of the microfibrils making a transversal cut. At a higher concentration, a greater reduction in the diameter of the FB is produced [44].

### 3.2. Size Particle Measurement by TEM

Figure 3 shows a TEM image used to identify that the CNCs are distributed from 24.4 nm to 97.5 nm in length and in diameter in a range of 2.2 nm to 18.6 nm, with an average length of 52.7 ± 18.1 nm and an average diameter of 7.5 ± 3.9 nm. This measurement is lower than the values generally reported in the process of obtaining cellulose nanoparticles using acid hydrolysis because, generally, lengths of 250 nm and diameters around 8 nm are reported, employing an acid solution at 64 wt.% [45]. The presence of the agglomerates can be attributed to van der Waals forces, which appear to a greater degree in particles with diameters under 25 µm, causing a greater attraction between them [46].

The CNC morphology is that of a slim rod with ends similar to the point of a needle. This structure might be associated with the detachment of the amorphous region of the cellulose, leaving the crystalline region exposed, which produces particles in slim rod form on a nanometric scale (see Figure 3) [41,43]. Other authors have obtained CNCs from sawdust waste using a sulphuric acid treatment with diameters of 35.2 nm and lengths of 238.7 nm with a rod form [47]. The morphology observed in the CNCs from fique bagasse is similar to that from other sources, such as sugarcane bagasse using acid hydrolysis, with lengths in a range of 200 to 300 nm and diameters of approximately 20 to 40 nm with a rod form. Similar data were reported for CNCs from pistachio using acid hydrolysis, with a 187 nm length and 12 nm diameter [48,49].

### 3.3. Structural Changes by FTIR of the FB in the Process of Obtaining Nanoparticles

To achieve nanometric-scale sizes, it was necessary to analyze the effect on the structure of the BC caused by the acid hydrolysis process with different concentrations of acid using FTIR. With this technique, the cellulose spectra were obtained to identify the structural changes of the cellulose of the FB in the process of obtaining nanoparticles. Figure 4 shows the spectra and a summary of the main bands of the treatments at different hydrolysis conditions, where peaks of greater intensity were observed with respect to the controls used, crude fique fiber (CF), crude bagasse, autoclaved crude bagasse (AFB), delignified crude bagasse (DFB), blanched crude bagasse (BFB), crude microcrystalline cellulose (CMC), and hydrolyzed microcrystalline cellulose (HMC) (see Figure 4). The wavelength oscillates between ±10 cm^−1^.

The 3335 cm^−1^ band is the characteristic peak of the stretching vibration of the OH group in the hydrogen bonds. With the hydrolysis process, this peak increased (see Figure 4) because this intensity is related to the increase in the cellulose content and the removal of the lignin fractions. Likewise, the 2916 cm^−1^ band corresponds to the asymmetric stretching of the lignin methyl group (CH_3_). The band from 2900 to 2850 cm^−1^ corresponds to the symmetric stretching vibration of the methylene group (=CH_2_) present in all spectra of the FB but with greatest intensity in the cellulose peaks. This band can show a displacement due to the NaOH concentration used in the delignification process. Similar data have been reported, finding that a high NaOH concentration generates a displacement of this band [50].

The peak of 1731 cm^−1^ represents the stretching vibration of the ester bond of the ferulic and p-coumaric acids of the lignin. A reduction in this vibration could be associated with the removal of the carboxyl groups in the FB components [51,52]. The 1618 cm^−1^ peak is associated with the bending vibration of the aromatic ring present in the lignin (C=C). This peak decreases its intensity on completion of the pretreatment prior to the hydrolysis. This may be due to the depolymerizing of the lignin during the pretreatment [53]. The 1240 cm^−1^ band corresponds to the C–O–C vibration of the aromatic group in the ester bond of lignin and hemicellulose. It was observed (see Figure 4) that these bands decrease in intensity and/or disappear after the treatments of delignification and blanching, which possibly indicates that these treatments were effective for the removal of these vibrations belonging to the lignin and hemicellulose groups [54].

Having completed the hydrolysis process, a structural change was observed in the non-cellulosic components of the BC in the 1710 and 1464 cm^−1^ bands (see Figure 5, Figure 6, Figure 7 and Figure 8) that correspond to the deformation and stretching of the COOH groups and the –CH bond, respectively. This behavior can be explained because the alkaline treatment does not completely remove the lignin through the presence of the C–C bonds and aromatic groups that are resistant to the chemical attack. Similar observations have been presented that report the resistance of these components to chemical treatments [55,56]. The peak reported in the 1644 cm^−1^ band in the spectra of MCC and MCH, meanwhile, can be attributed to the bending of the water absorbed in the cellulose [49,57].

The band located at 1427 cm^−1^ corresponds to the symmetric bending of the cellulose CH_2_ group, which is associated with the quantities of the crystalline zone of the cellulose. A decrement in this intensity reflects a reduction in the degree of crystallinity of the sample [58]. The symmetric bending of the cellulose methoxy group –CH bond is related to the 1368 cm^−1^ band, representing a decrease in its intensity due to the process of delignification and blanching of the FB [53]. The 1315 cm^−1^ band represents the symmetric bending of the –CH_2_ bond in carbon number 6 (C6) and the stretching of the cellulose S ring with a displacement to 1313 cm^−1^ that is related to the NaOH concentration (the higher the concentration, the greater the displacement), the development of new inter- and intramolecular hydrogen bonds, and a change in the conformation of the CH_2_OH in C6 [59].

Following the acid hydrolysis process, in the spectra of the different treatments (see Figure 5, Figure 6, Figure 7 and Figure 8), changes are observed in the structure of the FB with the appearance of new characteristic peaks of the functional groups of the cellulose. The 1201 cm^−1^ peak corresponds to the COH bending, the 1102 cm^−1^ peak is related to the symmetric vibration of the glucosidic ester C–O–C—mainly in the β-1,4-glucosidic bond—and the 1053 cm^−1^ peak is attributed to the stretching vibration of the pyranose ring C–O–C. These changes can be explained by the effectiveness of the acid treatment since it better exposes the functional groups [60]. The 1154 cm^−1^ band is present in all the spectra, and this is related to the stretching vibration of the pyranose skeletal ring C-C bonds present in the cellulose. These peaks and their variations in band number are influenced by the changes in the inter- and intramolecular hydrogen bonds, which are responsible for a number of properties of cellulose, lignin, and FB [50].

The presence of three peaks characteristic of cellulose was observed consistently in the spectra of the controls and the samples treated with the chemical methods (see Figure 5, Figure 6, Figure 7 and Figure 8). The stretching vibration of the C–O of C2, C3, and C6 corresponds to the 1028 cm^−1^ band. These vibrations were possibly due to the breaking of the glycosidic bond following the hydrolysis process [61]. The 897 cm^−1^ band is known as the amorphous zone of the cellulose and is assigned to the stretching C1–O–C4 in the β-1,4 glucosidic bonds. This band showed an increase in its intensity following the hydrolysis process because the amorphous zones were exposed [58]. Finally, following the delignification process (see Figure 5, Figure 6, Figure 7 and Figure 8), a change was observed in the 715 cm^−1^ band, which corresponds to the β cellulose produced by superior plants, due also to which the 750 cm^−1^ band is assigned to the α cellulose produced by primitive organisms, but this was not observed [62].

### 3.4. Effect of Temperature on Color Properties of Nanoparticles with Respect to Processing Time and Acid Concentration

The effect of temperature, processing time, and acid concentration on the nanoparticles shows significant differences. The lightness *L** presented values near 100 (100 = white color), but this value decreased when the temperature increased from 70 °C to 80 °C and 90 °C, showing different grouping levels for *L**. The chromatics coordinates *a** and *b** increased significantly with the increase in temperature, displacing the positive axis, indicated by the red and yellow colors, respectively. The temperature increase affects the different indices: it increased the chromaticity (C*), the yellowness index (*YI*), the saturation index (*SI*), the brownness index (*BI*), and the color index (CI), while the hue (h) and the whiteness index (*WI*) decreased (see Figure 9).

The color index in most of the samples gave values between −2 to +2, indicating a yellow-green color. However, three samples at a temperature of 90 °C turned a pale yellow color. The changes in color may be related to the oxidation of the glucose in its hydroxyl and carboxyl groups or in the residual lignin content. The lightest or darkest colors signal an increase or decrease in the lignin content of the particles [45]. Meanwhile, as mentioned earlier, temperature could also influence the color due to the fact that as this increases, darker particles are produced, an effect attributed to the oxidation of the products in the hemicellulose degradation process or migration of the lignin to the surface during the process of hydrolysis of the FB [45], a color difference (Δ*E***_ab_*) being found for the samples compared to commercial cellulose.

### 3.5. Mechanical Properties of the Bionanocomposite Matrix

Table 1 shows the results obtained in the measurement of the mechanical properties of the film with an inclusion of 0%, 5%, 10%, and 15% CNCs. The tensile stress showed (see Figure 10) an increase from 2.83 N to 4.88 N with 5% NPC added to the TPS film. The same behavior occurred with 10% CNC added, increasing from 2.83 N to 5.97 N for a total increase of 72.3% and 110.6%, respectively. Nevertheless, the behavior was not linear since, on addition of 15% CNC, the increase in the tensile stress rose from 2.83 N to just 5.73, less than that for the 10% CNC added. The CNC concentration has the greatest effect on the tensile stress property because of the entangling capacity of CNC in the material and due to its incapacity to retain different relations as regards a high CNC load. Moreover, the 15% added shows a limit to the CNC that can be added in the bionanocomposites and still achieve an increase in the tensile stress property [63].

Another cause reported for the loss in tensile stress is the CNC agglomeration rate that can break the matrix-fill interactions since the TPS tensile stress increase only occurs when there is sufficient matrix-fill forming a continuous structure and a suitable dispersion of CNCs. This behavior is found when one nanoparticle interacts with two or more, a phenomenon known as percolation [64].

Studies of bionanocomposites for the packaging of foods report that the increase in mechanical properties can be attributed to the high rigidity of the CNCs, as well as the excellent affinity between the biopolymer and the nanoparticles in the interphase [65]. Other authors report that this increase could be related to the high interfacial adhesion of the fill-matrix due to both the CNCs and the hydrophilic nature of the starch, as well as the fill–fill interactions [29,66]. A similar view on the increase in mechanical properties is shown for bionanocomposites obtained from starch reinforced with bacterial nanocrystal cellulose and pea starch reinforced with Lino nanocrystal cellulose [66]. Another explanation for this increase is attributed to the inter- and intramolecular hydrogen bonds and the good dispersion of the CNCs [48].

In the Young’s modulus, changes were observed (see Table 1) on addition of the different percentages of CNCs to the film. With an addition of 5% CNCs, there was an increase of 76.13%; with an addition of 10%, an increase of 219.85%, and for an addition of 15% CNCs, this change was 288.9% compared to TPS film with a CNC addition of 0%. Young’s modulus relates to the strength and deformation of a plastic material because this characteristic reveals the elastic behavior of the material. The higher this is, the less elastic the material and vice versa [29,30,67]. Accordingly, for materials made with jute and natural rubber cellulose nanofibers (CNFs), the tensile module increases due to the addition of CNFs in the polymeric matrix, reducing its mobility, and the composite material is more rigid with an increase in Young’s modulus; this increase is found with the highest percent addition of CNCs [50].

Other reports describe that Young’s modulus and elongation with a low nanocrystal cellulose (NCC) concentration (0.5 wt.%) produces no change. Nevertheless, some nanocomposites obtain an increase of approximately 8% in Young’s modulus, a behavior that could be related to the firmer segments in the microphase. In the film materials made with a blend of PVA and palm pulp nanocellulose, the Young’s modulus increased. The addition of NCCs decreased the deformation of both materials, bringing a reduction in the mobility of the segment and making the composite more rigid and firmer [68,69].

Finally, for elongation or percent deformation, which is an indicator of the flexibility and stretching capacity of a film, in the TPS film with a CNC addition of 0%, there was an elongation of 116.46%, a value that decreased with a CNC addition of 15% to a value of 75.23%. The same occurred with a CNC addition of 10%, decreasing to 97.14%. But with a CNC addition of 5%, elongation increased to a value of 146.22% (see Figure 10) because the addition of CNCs—through their hydrophilic nature, they can be water carriers—had a plasticizer effect on the polymeric matrix and caused an increase in ductility and thus an increase in elongation [70]. On the contrary, when the elongation decreases, this may be due to the addition of CNCs acting as a tension-concentrating component. Another cause is the rigid nature of the CNCs because these limit movement in the polymeric matrix due to the strong interaction between CNCs and the matrix. For rice films, elongation decreased from 53.6% to a minimum of 2.48% when the addition of CNCs increased from 5% to 30%, reporting that this decrease is possibly related to the differences in rigidity between the matrix and fillers because the greatest part of the deformation under high tension is coming from the polymer. Similar behavior is found in potato starch film with an addition of starch nanoparticles and with a decrease in deformation due to an agglomeration of CNCs within the films [70,71].

Figure 10 shows the TPS film to have a greater deformation than the TPS film with a CNC addition of 10% and 15%. This is possibly due to the effect of glycerol as a plasticizer since the intermolecular forces are reduced, as are the hydrogen bridges, rendering the material more flexible and less resistant [72]. Moreover, plasticizers also decrease the rate of crystallinity of the biopolymer films, reducing their strength and increasing their deformation [73]. Figure 11 shows the addition of 10% CNCs to the starch matrix film facilitated good mechanical properties, increasing the tensile stress by 110.71% and decreasing deformation by 16.59%. Therefore, it is possible that with this percentage addition of CNCs, there is an efficient dispersion of nanoparticles, enabling this starch film material to be considered as a promising packaging material. Additionally, for the bionanocomposite with 10% CNCs, the tensile stress was 5.97 N, while for commercial plastics, the tensile stress is 49.6 N, eight times as much. But, for the PHBV/Cloisite 30B Bionanocomposites, the tensile stress was 27 MPa and the elongation was 2.9%, and the elongation of the bionanocomposite with 5% CNCs was 116.95%, forty times as much; for a gelatin composite film with 7.5 wt.% CNCs, the tensile stress was 6.26 MPa, and the elongation was 14.5%, close values in reference to the bionanocomposite with fique bagasse CNPs.

### 3.6. Gas Oxygen Permeability of TPS Films with CNC Addition

The oxygen permeability test results are shown in Table 2, with no significant differences between the percentages of addition of CNCs (*p* < 0.05). Nevertheless, the addition of 10% CNCs led to a 20.46% decrease in the permeability of the TPS film. The addition of CNCs into the polymeric matrix allows an increase in the barrier properties in two ways: first, acting as a reinforcement, enhancing the compactness of the film due to reticulation between the polymer chains; secondly, reducing the mobility of the chains through the filling of free areas in the film matrix [74]. These quite possibly create a tortuous path such that the gas molecules are not able to easily penetrate the polymeric matrix film. The longer the diffusor path the molecules must take, the lower the permeability [29].

Another important reason to highlight is that the permeability is affected by the solubility, which is a characteristic related to the chemical nature, the crystallinity of the polymer, and the homogeneity of the sample. Therefore, the possibility exists that on using the casting method, a homogeneous dispersion of the CNCs could not be achieved, leading to the formation of agglomerations within the matrix film with a high CNC addition. The agglomeration may, in turn, lead to pores in the film, permitting more rapid penetration of the gas molecules through the matrix film [75]. Furthermore, the high deviation in the data can be associated with the casting method because the low homogeneity and the percentage of CNC addition in the matrix film have so many variations in the thickness of the samples in spite of preparing the samples using the same matrix film (see Figure 11). The reference value for the O_2_ gas permeability for the polyvinyl alcohol (PVA)/starch bionanocomposite with 5 wt.% halloysite nanotubes (HNT) made by the casting process was 0.97 × 10^−3^ g·m/m^2^·h·kPa.

### 3.7. Effect of the CNCs on the Structural Change in the Starch Film

Figure 12 shows SEM micrographs obtained for the cassava starch film adding 5%, 10%, and 15% CNCs at a magnification of 4000X on the surface and on the transverse section. From Figure 12, it is possible to identify the dispersion of the CNCs in the starch matrix film. This dispersion does not have a uniform, defined pattern, and the surface structure is similar in the starch films with each of the percentages of CNC addition. But in the transverse sections, a more compact material is observed for 5% and 10% addition of CNCs, while for 0% and 15%, the material has a porous structure, which means a loss in compacting material with the higher CNC addition of 15%; this shows that this change observed in the structure relates to loss in mechanical properties since the high addition of CNC facilitates the loss of interaction between the starch components and the CNCs [49].

### 3.8. Analysis of Thermal Properties of the CNCs and the Bionanocomposite Matrix

The thermogravimetric (TG) curves and the derivative thermogravimetric (DTG) curve are plotted in Figure 13A,B, respectively. The TG curves show a loss of weight at approximately 100 °C due to the evaporation of water from all sample surfaces [76]. Moreover, three heat-degradation processes were observed in the TG curves. The first, in the range 150 °C to 250 °C, presented depolymerization of the hemicellulose, having major activity in heat decomposition because of its random form in the chemical structure, and it hydrolyzes easily [50,52]. In the second, between 250 °C and 350 °C, the weight loss is possibly due to a group of endothermic reactions related to the depolymerization, dehydration, hydrolysis, oxidation, decarboxylation, or decomposition processes of the glucoside bond units in the cellulose where there is breaking of a bond such as –O–, C–H, C–O, or C–C, following the formation of carbonized residues [30,55,60]. The third and final weight loss in the TG curve is related to the slow decomposition of the lignin due to the production of unsaturated lateral chains that release water—CO_2_, CO, and CH_4_ [77].

In Figure 13B, the DTG curves show a lower decomposition temperature for DB, BB, and HB compared to FB and AB. This behavior may be associated with a higher lignin content in FB and AB. Lignin is composed of three benzene-propane types of units that are strongly reticulated and have a higher molecular weight, increasing their heat stability, so their heat degradation is a bit more difficult [75]. Another possible cause could be related to the high surface exposure to heat because the carboxylate groups present on the surface allow the direct phases to transition from solid to gas via decarboxylation [60] and the insertion of sulfate groups into the CNCs during the acid hydrolysis process [61,77]. The CNCs do not show any heat degradation in the temperature range of processing conventional polymers under 200 °C. These can therefore act as effective fill enhancement in biocomposites [78]. 

The TPS film weight loss on the addition of 0% and 10% in a temperature interval of 25 °C to 125 °C can be attributed to the evaporation of water weakly bonded to the surfaces of the film [47,66]. In the second step of degradation between 220 °C and 320 °C, a weight loss is generated that is related to heat degradation of the glycerol and the starch due to the removal of hydroxyl groups and the decomposition and depolymerization of the starch carbon chains [75]. The addition of the CNCs, meanwhile, also decreases the rate of decomposition in this temperature interval. A further aspect is that there are two peaks in the TPS film with a CNC addition of 10%. The first small peak may correspond to the dehydration of the cellulose, while the second peak could be due to the depolymerization of the cellulose in competition with the dehydration process [33].

The residual mass generated from 400 °C to 600 °C was highest for the TPS film with a CNC addition of 10% (13.3%) compared with the TPS film with a CNC addition of 0% (7.65%); this behavior is due to the high CNC heat stability, because in their composition there is a crystal with strong structures, and these cause a decrease in the polar character of the starch. The addition, therefore, of cellulose to the amylase matrix decreases the overall water content with an increase in the CNC content [75].

Fusion temperature (Tm) and heat of fusion (∆Hm) were measured by DSC analysis. The endothermic heat of decomposition is correlated with the crystallinity index as an indicator of the energy required to bring about the thermolysis of glycosidic bonds along the ordinate chains [41]. The increase in the endothermic heat flow for the TPS with a CNC addition of 10% is possibly related to the effect of agglomeration by the CNCs added, hindering mobility of the amorphous zones of the amylopectin chain, generating physical cross-linking induced by the growth of crystals. This showed that the strong interactions between the TPS and the CNCs increased the thermostability of the TPS films. Figure 13C shows that the fusion point of the TPS with the addition of 10% CNCs tends to decrease [79,80,81].

## 4. Conclusions

The temperature, time, and acid concentration effect in the hydrolysis process showed a statistically significant difference in obtaining CNCs, reducing the particle size of the FB, and achieving average sizes of 50.73 nm in length and 7.42 nm in diameter, with a rod shape and needle-like ends.

The pretreatments (delignification and bleaching) were shown to be adequate to carry out acid hydrolysis. This could be seen with the removal of some bands (1731 cm^−1^, 1618 cm^−1^, and 1240 cm^−1^) identified in the FTIR spectra corresponding to functional groups of lignin. And it was possible to identify the crystalline and amorphous zones of the cellulose, which were observed in the bands 1427 cm^−1^ (crystalline zone) and 897 cm^−1^ (amorphous zone) in the spectra of the hydrolyzed FB, indicating that the hydrolytic processes were effective.

Temperature is the most statistically significant difference factor in the determination of the CIEL*a*b* coordinates and the colorimetric indexes of the CNCs, obtaining lower tones and whiteness indexes at the temperature of 90 °C. Most of the particles of the different experimental treatments, meanwhile, had a greenish-yellow color index. 

The different levels of addition of CNCs in the TPS matrix film showed statistically significant differences in the mechanical tests. The 10% CNC inclusion saw favorable results in the film mechanical properties, increasing its strength by 110.71% and decreasing deformation by 16.59%, allowing it to be considered as a future packaging material. But, in the oxygen permeability with CNC inclusion in the TPS film, only 10% CNC inclusion showed a decrease of 20.46%, so its use as secondary packaging might be inferred.

## Figures and Tables

**Figure 1 polymers-15-04003-f001:**
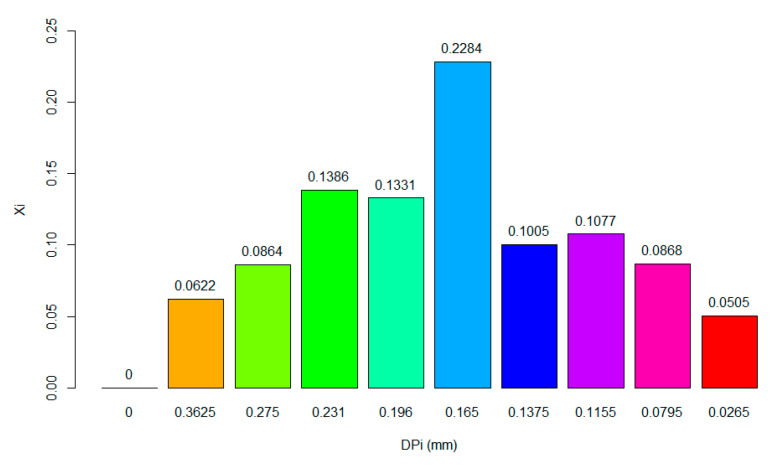
Particle size distribution of FB prior to the hydrolysis process.

**Figure 2 polymers-15-04003-f002:**
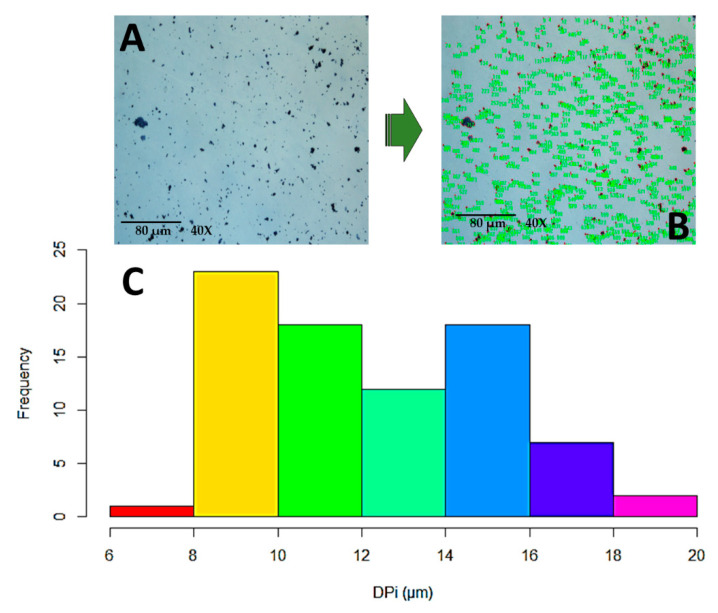
(**A**) Analysis of particle size of the hydrolyzed BC using images obtained with HROM. (**B**) Measurement of particle size from the HROM images using Image-Pro Plus Image Analysis and Processing Software, and (**C**) histogram of the hydrolyzed BC particle size distribution.

**Figure 3 polymers-15-04003-f003:**
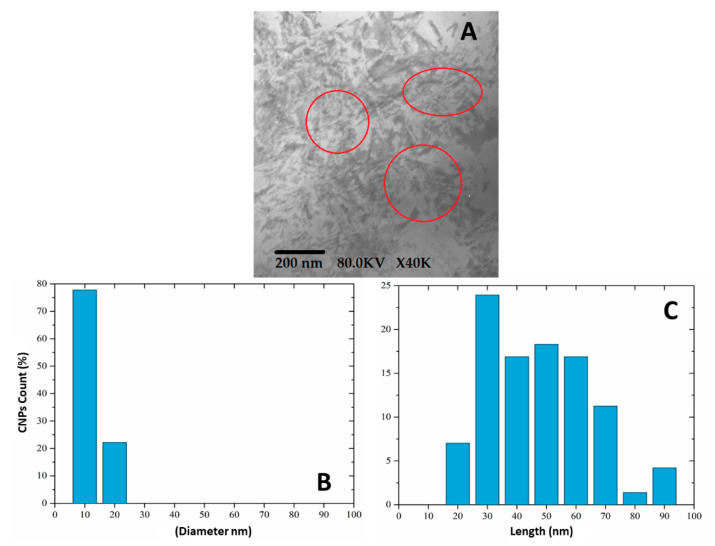
(**A**) TEM image of CNCs obtained by acid hydrolysis; size distribution of CNCs by their two dimensions, length and diameter, (**B**) and (**C**) respectively.

**Figure 4 polymers-15-04003-f004:**
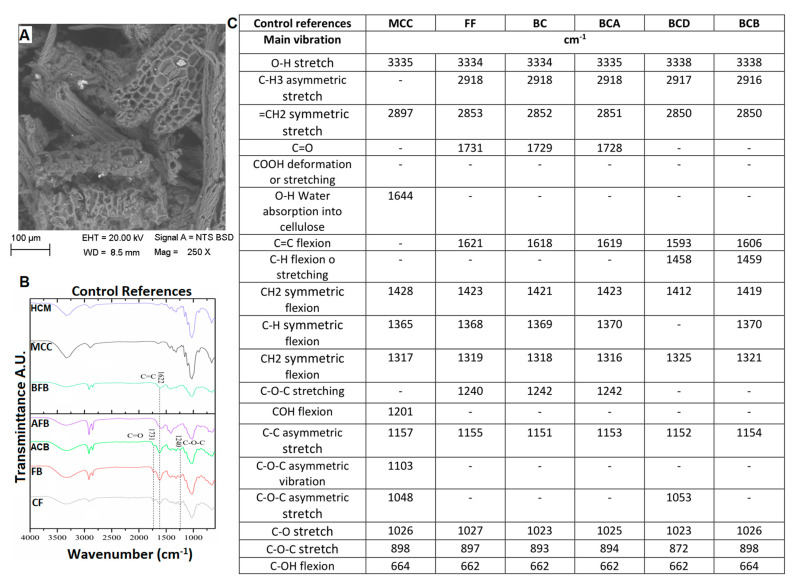
(**A**) FB SEM micrograph at 100 µm, 20 kV, and 250X. (**B**) FTIR spectra in the treatment process of the fique FB at 400 cm^−1^ to 4000 cm^−1^. (**C**) Summary of the main vibration of the samples (CF, FB, AFB, DFB, BFB, MCC, and HCM).

**Figure 5 polymers-15-04003-f005:**
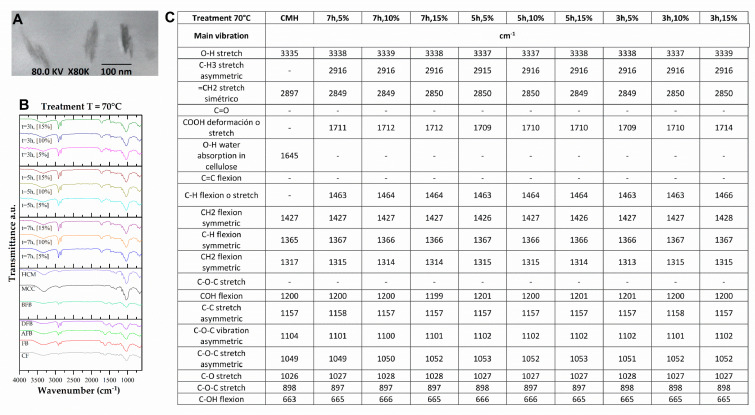
(**A**) HFB TEM micrograph at 100 nm, 80 kV, and 80 K X. (**B**) FTIR spectra in the treatment process of the fique FB at 400 cm^−1^ to 4000 cm^−1^. (**C**) Summary of the main vibration of the samples (CF, FB, AFB, DFB, BFB, MCC, HCM, and HBC at 70 °C).

**Figure 6 polymers-15-04003-f006:**
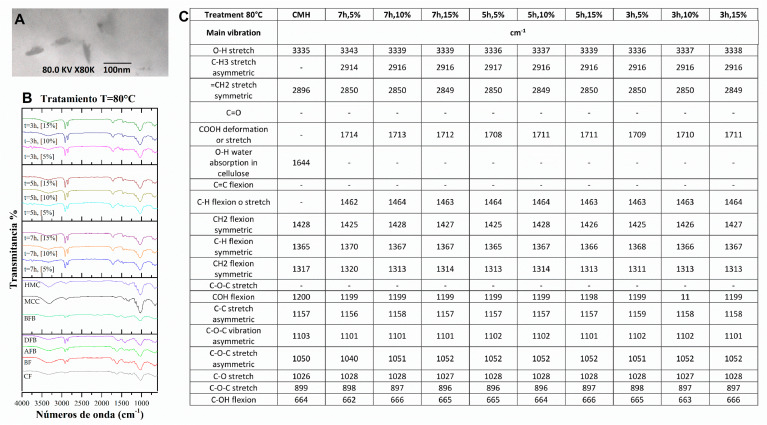
(**A**) HFB TEM micrograph at 100 nm, 80 kV, and 80 K X. (**B**) FTIR spectra in the treatment process of the fique FB. at 400 cm^−1^ to 4000 cm^−1^. (**C**) Summary of the main vibration of the samples (CF, FB, AFB, DFB, BFB, MCC, HCM, and HBC at 80 °C).

**Figure 7 polymers-15-04003-f007:**
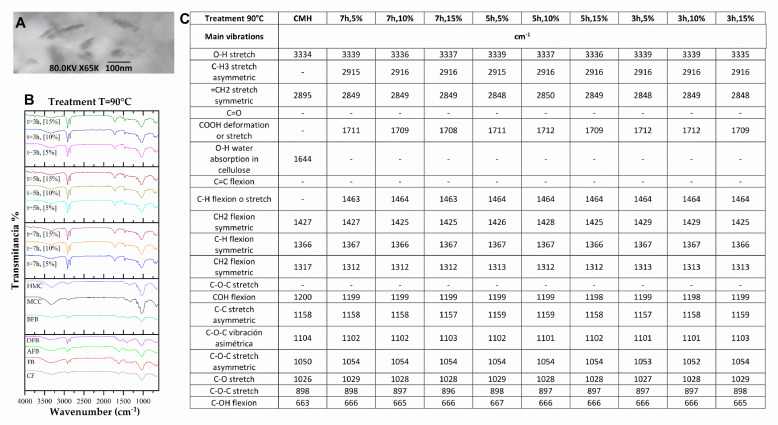
(**A**) HFB TEM micrograph at 100 nm, 80 kV, and 80 K X. (**B**) FTIR spectra in the treatment process of the fique FB at 400 cm^−1^ to 4000 cm^−1^. (**C**) Summary of the main vibration of the samples (CF, FB, AFB, DFB, BFB, MCC, HCM, and HFB at 90 °C).

**Figure 8 polymers-15-04003-f008:**
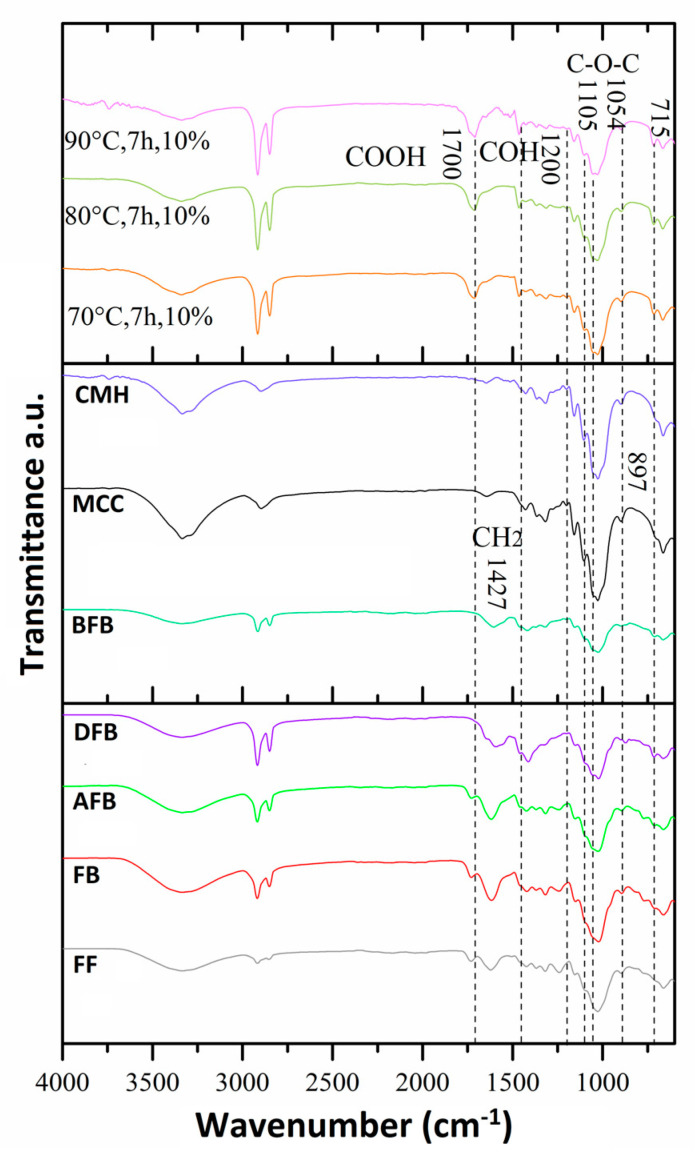
Comparison of the FTIR spectra of the best treatment (70 °C) with the others (80 °C and 90 °C and fique fiber (FF), crude bagasse (FB), autoclave crude bagasse (BCA), des lignin crude bagasse (BCD), blanching crude bagasse (BFB), microcrystalline cellulose (MCC), hydrolyzed microcrystalline cellulose, and FB under the acid hydrolysis process.

**Figure 9 polymers-15-04003-f009:**
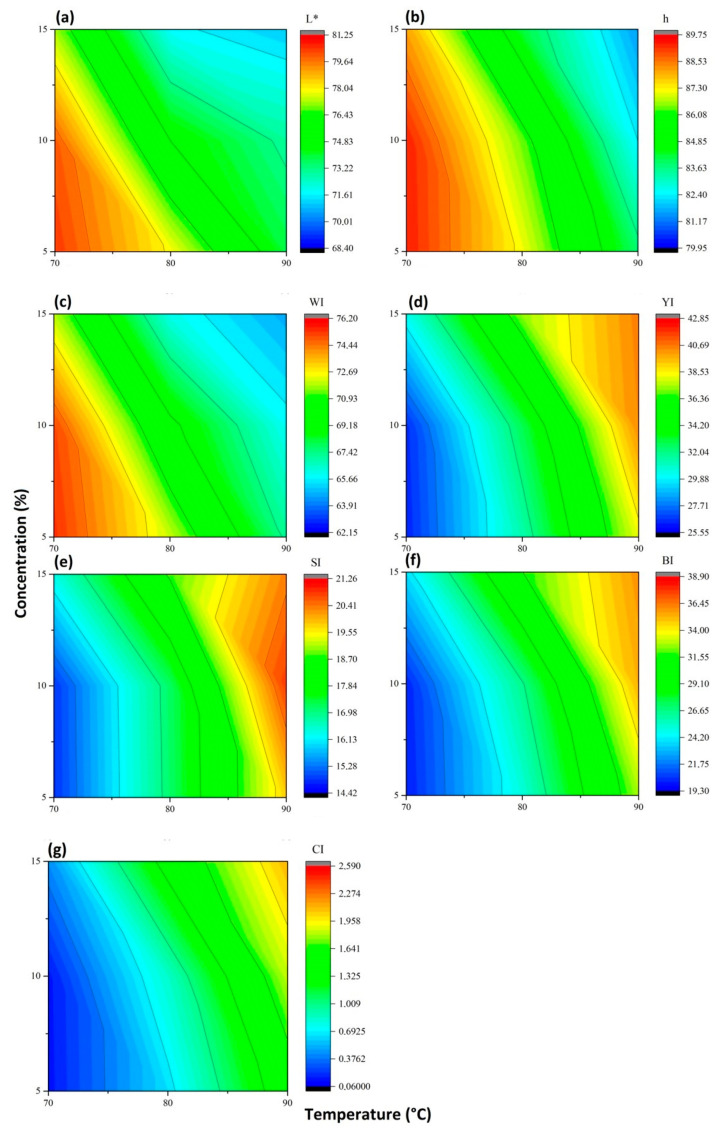
Correlation of the different indices with the temperature process and acid concentration. (**a**) Lightness; (**b**) hue; (**c**) whiteness index; (**d**) yellowness index; (**e**) saturation index; (**f**) brownness index; and (**g**) color index with a D65 light source, an angle of observation of 10°, and a calibration standard Z (89.5); X (0.3176); Y (0.3347).

**Figure 10 polymers-15-04003-f010:**
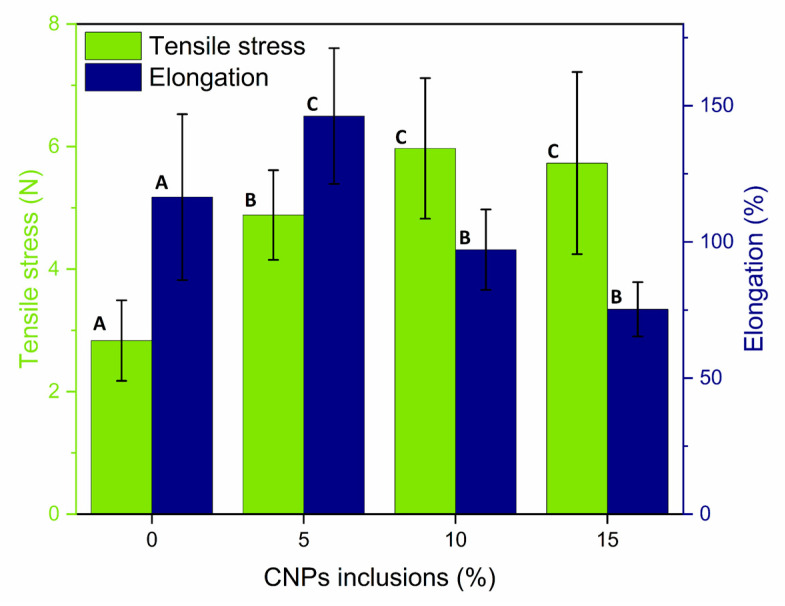
Effect of inclusion of CNCs on the tensile stress and the elongation of the starch matrix (bionanocomposite). Different letters mean a significant difference (*p* < 0.05).

**Figure 11 polymers-15-04003-f011:**
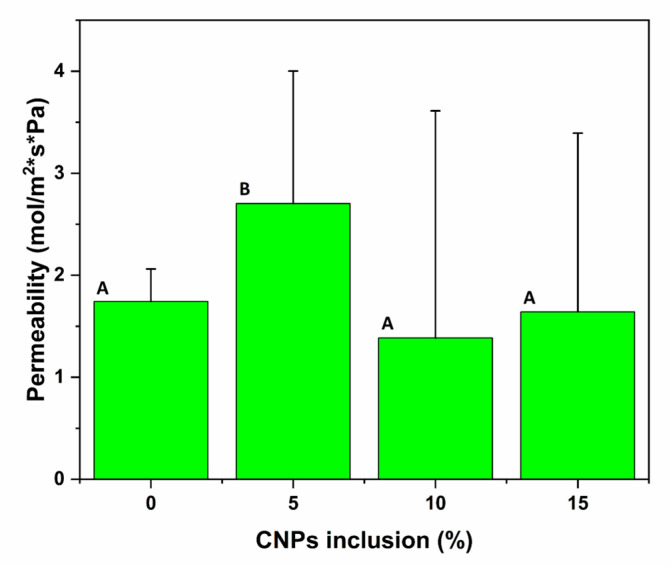
Effect of addition of CNCs in the TPS film on oxygen gas permeability for 24 h and 14 psi. The different letters mean significant differences (*p* < 0.05).

**Figure 12 polymers-15-04003-f012:**
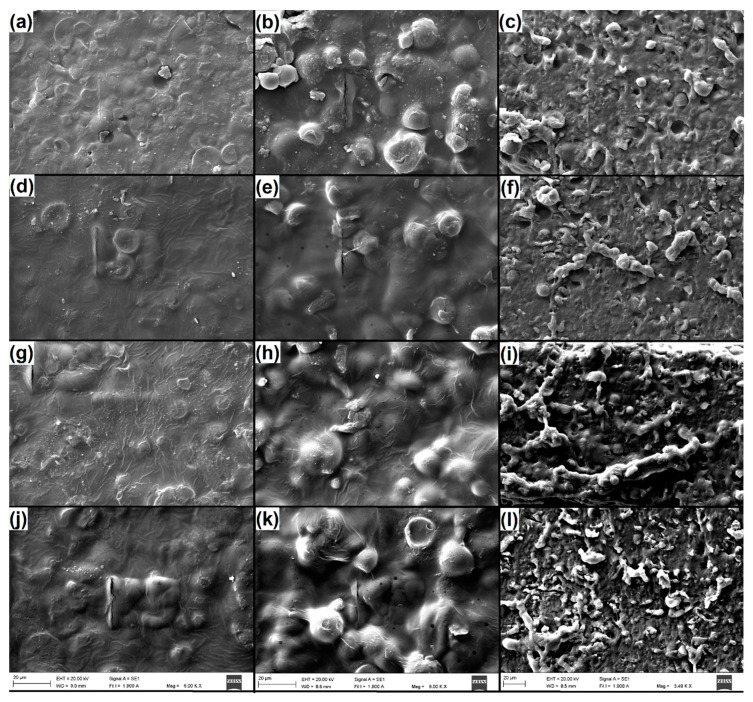
Starch film structure. (**a**–**c**) TPS 0% addition of CNCs. (**d**–**f**) TPS 5% addition of CNCs. (**g**–**i**) TPS 10% addition of CNCs. (**j**–**l**) TPS 15% addition of CNCs. SEM micrographs at 20 kV and 20 µm.

**Figure 13 polymers-15-04003-f013:**
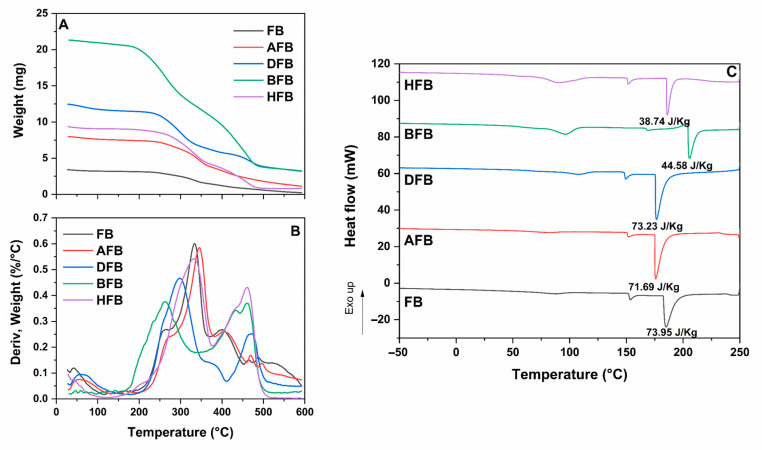
(**A**) TGA of FB in the hydrolysis process and (**B**) DTGA of FB in the hydrolysis process. (**C**) Thermal behavior of CNCs and the bionanocomposite matrix obtained by DSC.

**Table 1 polymers-15-04003-t001:** Average values of the TPS mechanical properties with and without CNC addition.

Sample	Tensile Modulus	Tensile Stress	Elongation
	(MPa)	(N)	(%)
TPS	0.8 ± 0.2	2.8 ± 0.7	116.5 ± 30.5
TPS + 5%NPC	1.4 ± 0.4	4.9 ± 0.7	117.0 ± 25.0
TPS + 10%NPC	2.6 ± 0.5	6.0 ± 0.8	97.1 ± 29.0
TPS + 15%NPC	3.1 ± 0.9	5.7 ± 1.5	75.2 ± 10.0

**Table 2 polymers-15-04003-t002:** Gas oxygen permeability of TPS films with different percentages of addition of CNCs.

Sample	GTR (mol/m^2^s)	Permeability (mol/m^2^sPa)
0%	1.77 × 10^−6^ ± 3.25 × 10^−7^	1.74 × 10^−11^ ± 3.20 × 10^−12^
5%	2.74 × 10^−6^ ± 1.32 × 10^−6^	2.70 × 10^−11^ ± 1.30 × 10^−11^
10%	1.40 × 10^−6^ ± 2.25 × 10^−6^	1.39 × 10^−11^ ± 2.23 × 10^−11^
15%	7.17 × 10^−7^ ± 6.13 × 10^−7^	2.59 × 10^−11^ ± 1.75 × 10^−11^

## Data Availability

The data presented in this study are available on request from the corresponding author.

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
