# Peer review of "Effect of the Addition of Fique Bagasse Cellulose Nanoparticles on the Mechanical and Structural Properties of Plastic Flexible Films from Cassava Starch"

_polymers, 2023, doi:10.3390/polym15194003_

Round 1

Reviewer 1 Report

The authors prepared cellulose nanomaterials which were used to reinforce starch films. This kind of work has been extensively studied. The novelty of this work maybe focus on the use of fique bagasse. Detailed comments are given below.

1) Cellulose nanoparticles are not specific. According to the method used for preparation of cellulose nanoparticles, this CNP should be cellulose nanocrystals (CNC) based on the definition of ISO standard. So please correct throughout the manuscript. 

2) Page 1, lines 29-30, what do you mean about "its use period is approximately just 12-20 minutes"? I don't agree with this description.

3) for the unit of time, like minutes and hours, please use their abbreviations min and h, respectively. please check and revise throughout the manuscript, to be consistent.

4) what's the scale bar for figure 2A and 2B?

5) The quality of the TEM of CNC in Figure 3A is too low. The CNC particles are not well dispersed. many aggregates. Also, the length is shorter than the diameter, according to figure 3B and 3C? please correct.

6) The quality of Figures 5A, 5B, 6A, 6B, 7A, and 7B is too low.

7) In Figure 10, which one is A, and which one is B? The caption needs to be revised.

8) The strength and barrier properties of the composite film should be compared with commercial plastics and similar films reported in literatures, to see whether the composite film is suitable for practical use.

9) Conclusion needs to be revised and well summarized.

Minor revision of English writing is needed.

Author Response

The authors present to you the respective corrections did it to the manuscript.

We will be attentive to your answer.

Bets regards,

Reviewer 2 Report

The manuscript of Jhon Jairo Palechor-Trochez et al is devoted to the production of composites based on cellulose obtained from fique fiber, which are obtained in large quantities when grown in Colombia. Thus, the considered source of cellulose has a sufficient raw material base, which is currently partially used. The authors consider various conditions for the extraction of cellulose. The resulting cellulose is introduced into the cassava starch matrix. The authors have deeply worked on this problem, so the list of references covers 118 sources. Unfortunately, the manuscript contains a large number of typos, errors, etc. Hence, it is necessary to work on correcting these errors. The quality of some drawings needs to be improved. In my opinion, the manuscript is more like a Materials magazine than Polymers. Therefore, I recommend that authors consider publishing in Materials. The novelty of the work lies only in the very source of cellulose.   L. 115. "NPCs" -?! L.94. "obtention" - needs to be fixed. L.96. "51]y". L.180. Replace "Evaluation of Tensile Mechanical Properties" with "Evaluation of Mechanical Properties" Figure 2. You need to add a scale bar to the photo. Figure 4. The quality of the IR spectra is very poor. L.142, 143, 326 etc. "attenuated total reflectance (ATR)" - "transmittance,%" - ?! Table 1. It is required to round all the values presented in the table, especially for elongation! Figure 12. Need to improve the quality of the scale bar. L.535. "." - delete. Figure 13B. It follows from the graph that the mass of the samples is increasing?! Reference 75 needs to be corrected.

Author Response

The authors present to you the respective corrections did it to the manuscript.

We will be attentive to your answer.

Bets regards

Round 2

Reviewer 1 Report

The manuscript has been improved. But there are too many references (totally 118) cited in this manuscript. Please reduce the number of references.

Author Response

The authors present to yo the corrections did it to manuscript.

Reviewer 2 Report

It is necessary to edit the text of the manuscript and correct stylistic and grammatical errors. Keywords: proposal to replace lignocellulosic with lignocellulosic biomass L.87. "crude" I propose to remove and change the abbreviation to FB. L.141, 142. "transmittance in percentage (%)" ATR- Attenuated total reflectance - it is necessary to check and correct the terminology used! L.180. "mechanical" - can be deleted! L.341. "Figs" - needs to be fixed. Table 1. It is necessary to round the values!!! "The interaction of the three factors of temperature, time, and acid" -?! L.581. "blanching" perhaps the author meant "bleaching" Conclusion. - It is necessary to check the text!

Author Response

The authors present to you the correction did it to the manuscript.
